# Analyzing Hogwild Parallel Gaussian Gibbs Sampling

**Matthew J. Johnson**
EECS, MIT
mattjj@mit.edu

**James Saunderson**
EECS, MIT
jamess@mit.edu

**Alan S. Willsky**
EECS, MIT
willsky@mit.edu

## Abstract

Sampling inference methods are computationally difficult to scale for many models in part because global dependencies can reduce opportunities for parallel computation. Without strict conditional independence structure among variables, standard Gibbs sampling theory requires sample updates to be performed sequentially, even if dependence between most variables is not strong. Empirical work has shown that some models can be sampled effectively by going "Hogwild" and simply running Gibbs updates in parallel with only periodic global communication, but the successes and limitations of such a strategy are not well understood.

As a step towards such an understanding, we study the Hogwild Gibbs sampling strategy in the context of Gaussian distributions. We develop a framework which provides convergence conditions and error bounds along with simple proofs and connections to methods in numerical linear algebra. In particular, we show that if the Gaussian precision matrix is generalized diagonally dominant, then any Hogwild Gibbs sampler, with any update schedule or allocation of variables to processors, yields a stable sampling process with the correct sample mean.

## 1 Introduction

Scaling probabilistic inference algorithms to large datasets and parallel computing architectures is a challenge of great importance and considerable current research interest, and great strides have been made in designing parallelizeable algorithms. Along with the powerful and sometimes complex new algorithms, a very simple strategy has proven to be surprisingly successful in some situations: running Gibbs sampling updates, derived only for the sequential setting, in parallel without globally synchronizing the sampler state after each update. Concretely, the strategy is to apply an algorithm like Algorithm 1. We refer to this strategy as "Hogwild Gibbs sampling" in reference to recent work [1] in which sequential computations for computing gradient steps were applied in parallel (without global coordination) to great beneficial effect.

This Hogwild Gibbs sampling strategy has long been considered a useful hack, perhaps for preparing decent initial states for a proper serial Gibbs sampler, but extensive empirical work on Approximate Distributed Latent Dirichlet Allocation (AD-LDA) [2, 3, 4, 5, 6], which applies the strategy to generate samples from a collapsed LDA model, has demonstrated its effectiveness in sampling LDA models with the same predictive performance as those generated by standard serial Gibbs [2, Figure 3]. However, the results are largely empirical and so it is difficult to understand how model properties and algorithm parameters might affect performance, or whether similar success can be expected for any other models. There have been recent advances in understanding some of the particular structure of AD-LDA [6], but a thorough theoretical explanation for the effectiveness and limitations of Hogwild Gibbs sampling is far from complete.

Sampling inference algorithms for complex Bayesian models have notoriously resisted theoretical analysis, so to begin an analysis of Hogwild Gibbs sampling we consider a restricted class of models that is especially tractable for analysis: Gaussians. Gaussian distributions and algorithms are tractable because of their deep connection with linear algebra. Further, Gaussian sampling is of

**Algorithm 1** Hogwild Gibbs Sampling

---

**Require:** Samplers $G_i(\bar{x}_{\neg i})$ which sample $p(x_i | x_{\neg i} = \bar{x}_{\neg i})$, a partition $\{\mathcal{I}_1, \mathcal{I}_2, \ldots, \mathcal{I}_K\}$ of $\{1, 2, \ldots, n\}$, and an inner iteration schedule $q(k, \ell) \geq 0$

1: Initialize $\bar{x}^{(1)}$
2: **for** $\ell = 1, 2, \ldots$ until convergence **do**                    $\triangleright$ global iterations/synchronizations
3:     **for** $k = 1, 2, \ldots, K$ in parallel **do**                    $\triangleright$ for each of $K$ parallel processors
4:         $\bar{y}_{\mathcal{I}_k}^{(1)} \leftarrow \bar{x}_{\mathcal{I}_k}^{\ell}$
5:         **for** $j = 1, 2, \ldots, q(k, \ell)$ **do**                    $\triangleright$ run local Gibbs steps with old
6:             **for** $i \in \mathcal{I}_k$ **do**                    $\triangleright$ statistics from other processors
7:                 $\bar{y}_i^{(j)} \leftarrow G_i(\bar{x}_{\mathcal{I}_1}^{(\ell)}, \ldots, \bar{y}_{\mathcal{I}_k \setminus \{i\}}^{(j)}, \ldots, \bar{x}_{\mathcal{I}_K}^{(\ell)})$
8:     $\bar{x}^{(\ell+1)} \leftarrow (\bar{y}_{\mathcal{I}_1}^{(q(1,\ell))} \cdots \bar{y}_{\mathcal{I}_K}^{(q(K,\ell))})$                    $\triangleright$ globally synchronize statistics

---

great interest in its own right, and there is active research in developing powerful Gaussian samplers [7, 8, 9, 10]. Gaussian Hogwild Gibbs sampling could be used in conjunction with those methods to allow greater parallelization and scalability, given an understanding of its applicability and tradeoffs.

Toward the goal of understanding Gaussian Hogwild Gibbs sampling, the main contribution of this paper is a linear algebraic framework for analyzing the stability and errors in Gaussian Hogwild Gibbs sampling. Our framework yields several results, including a simple proof for a sufficient condition for all Gaussian Hogwild Gibbs sampling processes to be stable and yield the correct asymptotic mean no matter the allocation of variables to processors or number of sub-iterations (Proposition 1, Theorem 1), as well as an analysis of errors introduced in the process variance.

Code to regenerate our plots is available at `https://github.com/mattjj/gaussian-hogwild-gibbs`.

## 2 Related Work

There has been significant work on constructing parallel Gibbs sampling algorithms, and the contributions are too numerous to list here. One recent body of work [11] provides exact parallel Gibbs samplers which exploit graphical model structure for parallelism. The algorithms are supported by the standard Gibbs sampling analysis, and the authors point out that while heuristic parallel samplers such as the AD-LDA sampler offer easier implementation and often greater parallelism, they are currently not supported by much theoretical analysis.

The parallel sampling work that is most relevant to the proposed Hogwild Gibbs sampling analysis is the thorough empirical demonstration of AD-LDA [2, 3, 4, 5, 6] and its extensions. The AD-LDA sampling algorithm is an instance of the strategy we have named Hogwild Gibbs, and Bekkerman et al. [5, Chapter 11] suggests applying the strategy to other latent variable models.

The work of Ihler et al. [6] provides some understanding of the effectiveness of a variant of AD-LDA by bounding in terms of run-time quantities the one-step error probability induced by proceeding with sampling steps in parallel, thereby allowing an AD-LDA user to inspect the computed error bound after inference [6, Section 4.2]. In experiments, the authors empirically demonstrate very small upper bounds on these one-step error probabilities, e.g. a value of their parameter $\varepsilon = 10^{-4}$ meaning that at least $99.99\%$ of samples are expected to be drawn just as if they were sampled sequentially. However, this per-sample error does not necessarily provide a direct understanding of the effectiveness of the overall algorithm because errors might accumulate over sampling steps; indeed, understanding this potential error accumulation is of critical importance in iterative systems. Furthermore, the bound is in terms of empirical run-time quantities, and thus it does not provide guidance regarding on which other models the Hogwild strategy may be effective. Ihler et al. [6, Section 4.3] also provides approximate scaling analysis by estimating the order of the one-step bound in terms of a Gaussian approximation and some distributional assumptions.

Finally, Niu et al. [1] provides both a motivation for Hogwild Gibbs sampling as well as the Hogwild name. The authors present "a lock-free approach to parallelizing stochastic gradient descent" (SGD) by providing analysis that shows, for certain common problem structures, that the locking

and synchronization needed to run a stochastic gradient descent algorithm "correctly" on a multicore architecture are unnecessary, and in fact the robustness of the SGD algorithm compensates for the uncertainty introduced by allowing processors to perform updates without locking their shared memory.

## 3 Background

In this section we fix notation for Gaussian distributions and describe known connections between Gaussian sampling and a class of stationary iterative linear system solvers which are useful in analyzing the behavior of Hogwild Gibbs sampling.

The density of a Gaussian distribution on $n$ variables with mean vector $\mu$ and positive definite[1] covariance matrix $\Sigma \succ 0$ has the form

$$p(x) \propto \exp\left\{-\tfrac{1}{2}(x-\mu)^\mathsf{T}\Sigma^{-1}(x-\mu)\right\} \propto \exp\left\{-\tfrac{1}{2}x^\mathsf{T}Jx + h^\mathsf{T}x\right\} \tag{1}$$

where we have written the information parameters $J := \Sigma^{-1}$ and $h := J\mu$. The matrix $J$ is often called the *precision matrix* or *information matrix*, and it has a natural interpretation in the context of Gaussian graphical models: its entries are the coefficients on pairwise log potentials and its sparsity pattern is exactly the sparsity pattern of a graphical model. Similarly $h$, also called the *potential vector*, encodes node potentials and evidence.

In many problems [12] one has access to the pair $(J, h)$ and must compute or estimate the moment parameters $\mu$ and $\Sigma$ (or just the diagonal) or generate samples from $\mathcal{N}(\mu, \Sigma)$. Sampling provides both a means for estimating the moment parameters and a subroutine for other algorithms. Computing $\mu$ from $(J, h)$ is equivalent to solving the linear system $J\mu = h$ for $\mu$.

One way to generate samples is via Gibbs sampling, in which one iterates sampling each $x_i$ conditioned on all other variables to construct a Markov chain for which the invariant distribution is the target $\mathcal{N}(\mu, \Sigma)$. The conditional distributions for Gibbs sampling steps are $p(x_i|x_{\neg i} = \bar{x}_{\neg i}) \propto \exp\left\{-\tfrac{1}{2}J_{ii}x_i^2 + (h_i - J_{i\neg i}\bar{x}_{\neg i})x_i\right\}$. That is, we update each $x_i$ via $x_i \leftarrow \frac{1}{J_{ii}}(h_i - J_{i\neg i}\bar{x}_{\neg i}) + v_i$ where $v_i \overset{\text{iid}}{\sim} \mathcal{N}(0, \frac{1}{J_{ii}})$.

Since each variable update is a linear function of other variables with added Gaussian noise, we can collect one scan for $i = 1, 2, \ldots, n$ into a matrix equation relating the sampler state at $t$ and $t + 1$:

$$x^{(t+1)} = -D^{-1}Lx^{(t+1)} - D^{-1}L^\mathsf{T}x^{(t)} + D^{-1}h + D^{-\frac{1}{2}}\tilde{v}^{(t)}, \quad \tilde{v}^{(t)} \overset{\text{iid}}{\sim} \mathcal{N}(0, I).$$

where we have split $J = L + D + L^\mathsf{T}$ into its strictly lower-triangular, diagonal, and strictly upper-triangular parts, respectively. Note that $x^{(t+1)}$ appears on both sides of the equation, and that the sparsity patterns of $L$ and $L^\mathsf{T}$ ensure that each entry of $x^{(t+1)}$ depends on the appropriate entries of $x^{(t)}$ and $x^{(t+1)}$. We can re-arrange the equation into an update expression:

$$x^{(t+1)} = -(D+L)^{-1}L^\mathsf{T}x^{(t)} + (D+L)^{-1}h + (D+L)^{-1}\tilde{v}^{(t)}, \quad \tilde{v}^{(t)} \overset{\text{iid}}{\sim} \mathcal{N}(0, D).$$

The expectation of this update is exactly the Gauss-Seidel iterative linear system solver update [13, Section 7.3] applied to $J\mu = h$, i.e. $x^{(t+1)} = -(D+L)^{-1}L^\mathsf{T}x^{(t)} + (D+L)^{-1}h$. Therefore a Gaussian Gibbs sampling process can be interpreted as Gauss-Seidel iterates on the system $J\mu = h$ with appropriately-shaped noise injected at each iteration.

Gauss-Seidel is one instance of a stationary iterative linear solver based on a *matrix splitting*. In general, one can construct a stationary iterative linear solver for any splitting $J = M - N$ where $M$ is invertible, and similarly one can construct iterative Gaussian samplers via

$$x^{(t+1)} = (M^{-1}N)x^{(t)} + M^{-1}h + M^{-1}v^{(t)}, \quad v^{(t)} \overset{\text{iid}}{\sim} \mathcal{N}(0, M^\mathsf{T} + N) \tag{2}$$

with the constraint that $M^\mathsf{T} + N \succeq 0$ (i.e. that the splitting is P-regular [14]). For an iterative process like (2) to be *stable* or *convergent* for any initialization we require the eigenvalues of its

update map to lie in the interior of the complex unit disk, i.e. $\rho(M^{-1}N) := \max_i |\lambda_i(M^{-1}N)| < 1$ [13, Lemma 7.3.6]. The Gauss-Seidel solver (and Gibbs sampling) correspond to choosing $M$ to be the lower-triangular part of $J$ and $N$ to be the negative of the strict upper-triangle of $J$. $J \succeq 0$ is a sufficient condition for Gauss-Seidel to be convergent [13, Theorem 7.5.41] [15], and the connection to Gibbs sampling provides an independent proof.

For solving linear systems with splitting-based algorithms, the complexity of solving linear systems in $M$ directly affects the computational cost per iteration. For the Gauss-Seidel splitting (and hence Gibbs sampling), $M$ is chosen to be lower-triangular so that the corresponding linear system can be solved efficiently via backsubstitution. In the sampling context, the per-iteration computational complexity is also determined by the covariance of the injected noise process $v^{(t)}$, because at each iteration one must sample from a Gaussian distribution with covariance $M^{\mathsf{T}} + N$.

We highlight one other standard stationary iterative linear solver that is relevant to analyzing Gaussian Hogwild Gibbs sampling: Jacobi iterations, in which one splits $J = D - A$ where $D$ is the diagonal part of $J$ and $A$ is the negative of the off-diagonal part. Due to the choice of a diagonal $M$, each coordinate update depends only on the previous sweep's output, and thus the Jacobi update sweep can be performed in parallel. A sufficient condition for the convergence of Jacobi iterates is for $J$ to be a generalized diagonally dominant matrix (i.e. an H-matrix) [13, Definition 5.13]. A simple proof [2] due to Ruozzi et al. [16], is to consider Gauss-Seidel iterations on a *lifted* $2n \times 2n$ system:

$$\begin{pmatrix} D & -A \\ -A & D \end{pmatrix} \xrightarrow{\text{G-S update}} \begin{pmatrix} D^{-1} & 0 \\ D^{-1}AD^{-1} & D^{-1} \end{pmatrix} \begin{pmatrix} 0 & A \\ 0 & 0 \end{pmatrix} = \begin{pmatrix} 0 & D^{-1}A \\ 0 & (D^{-1}A)^2 \end{pmatrix}. \tag{3}$$

Therefore one iteration of Gauss-Seidel on the lifted system is exactly two applications of the Jacobi update $D^{-1}A$ to the second half of the state vector, so Jacobi iterations converge if Gauss-Seidel on the lifted system converges. Furthermore, a sufficient condition for Gauss-Seidel to converge on the lifted system is for it to be positive semi-definite, and by taking Schur complements we require $D - AD^{-1}A \succeq 0$ or $I - (D^{-\frac{1}{2}}AD^{-\frac{1}{2}})(D^{-\frac{1}{2}}AD^{-\frac{1}{2}}) \succeq 0$, which is equivalent to requiring generalized diagonal dominance [13, Theorem 5.14].

## 4 Gaussian Hogwild Analysis

Given that Gibbs sampling iterations and Jacobi solver iterations, which can be computed in parallel, can each be written as iterations of a stochastic linear dynamical system (LDS), it is not surprising that Gaussian Hogwild Gibbs sampling can also be expressed as an LDS by appropriately composing these ideas. In this section we describe the LDS corresponding to Gaussian Hogwild Gibbs sampling and provide convergence and error analysis, along with a connection to a class of linear solvers.

For the majority of this section, we assume that the number of inner iterations performed on each processor is constant across time and processor index; that is, we have a single number $q = q(k, \ell)$ of sub-iterations per processor for each outer iteration. We describe how to relax the assumption at the end of this subsection.

Given a joint Gaussian distribution of dimension $n$ represented by a pair $(J, h)$ as in (1), we represent an allocation of the $n$ scalar variables to local processors by a partition of $\{1, 2, \ldots, n\}$, where we assume partition elements are contiguous without loss of generality. Consider a block-Jacobi splitting of $J$ into its block diagonal and block off-diagonal components, $J = D_{\text{block}} - A$, according to the partition. $A$ includes the cross-processor potentials, and this block-Jacobi splitting will model the outer iterations in Algorithm 1. We further perform a Gauss-Seidel splitting on $D_{\text{block}}$ into (block-diagonal) lower-triangular and strictly upper-triangular parts, $D_{\text{block}} = B - C$; these processor-local Gauss-Seidel splittings model the inner Gibbs sampling steps in Algorithm 1. We refer to this splitting $J = B - C - A$ as the Hogwild splitting; see Figure 1a for an example.

For each outer iteration of the Hogwild Gibbs sampler we perform $q$ processor-local Gibbs steps, effectively applying the block-diagonal update $B^{-1}C$ repeatedly using $Ax^{(t)} + h$ as a potential

vector that includes out-of-date statistics from the other processors. The resulting update operator for one outer iteration of the Hogwild Gibbs sampling process can be written as

$$x^{(t+1)} = (B^{-1}C)^q x^{(t)} + \sum_{j=0}^{q-1} (B^{-1}C)^j B^{-1} \left( Ax^{(t)} + h + v^{(t,j)} \right), \quad v^{(t,j)} \overset{\text{iid}}{\sim} \mathcal{N}(0, D) \quad (4)$$

where $D$ is the diagonal of $J$. Note that we shape the noise diagonally because in Hogwild Gibbs sampling we simply apply standard Gibbs updates in the inner loop.

As mentioned previously, the update in (4) is written so that the number of sub-iterations is homogeneous, but the expression can easily be adapted to model any numbers of sub-iterations by writing a separate sum over $j$ for each block row of the output and a separate matrix power for each block in the first $B^{-1}C$ term. The proofs and arguments in the following subsections can also be extended with extra bookkeeping, so we focus on the homogeneous $q$ case for convenience.

## 4.1 Convergence and Correctness of Means

Because the Gaussian Hogwild Gibbs sampling iterates form a Gaussian linear dynamical system, the process is stable (i.e. its iterates converge in distribution) if and only if [13, Lemma 7.3.6] the deterministic part of the update map (4) has spectral radius less than unity, i.e.

$$T := (B^{-1}C)^q + \sum_{j=0}^{q-1} (B^{-1}C)^j B^{-1} A = (B^{-1}C)^q + (I - (B^{-1}C)^q)(B - C)^{-1} A \quad (5)$$

satisfies $\rho(T) < 1$. We can write $T = T_{\text{ind}}^q + (I - T_{\text{ind}}^q)T_{\text{block}}$ where $T_{\text{ind}}$ is the purely Gauss-Seidel update when $A = 0$ and $T_{\text{block}}$ for the block Jacobi update, which corresponds to solving the processor-local linear systems exactly at each outer iteration. The update (5) falls into the class of two-stage splitting methods [14, 17, 18], and the next proposition is equivalent to such two-stage solvers having the correct fixed point.

**Proposition 1.** *If a Gaussian Hogwild Gibbs process is stable, the asymptotic mean is correct.*

*Proof.* If the process is stable the mean process has a unique fixed point, and from (4) and (5) we can write the fixed-point equation for the process mean $\mu_{\text{hog}}$ as $(I-T)\mu_{\text{hog}} = (I-T_{\text{ind}})(I-T_{\text{block}})\mu_{\text{hog}} = (I-T_{\text{ind}})(B-C)^{-1}h$, hence $(I-(B-C)^{-1}A)\mu_{\text{hog}} = (B-C)^{-1}h$ and $\mu_{\text{hog}} = (B-C-A)^{-1}h$. $\square$

The behavior of the spectral radius of the update map can be very complicated, even generically over simple ensembles. In Figure 1b, we compare $\rho(T)$ for $q = 1$ and $q = \infty$ (corresponding to $T = T_{\text{block}}$) with models sampled from a natural random ensemble; we see that there is no general relationship between stability at $q = 1$ and at $q = \infty$.

Despite the complexity of the update map's stability, in the next subsection we give a simple argument that identifies its convergence with the convergence of Gauss-Seidel iterates on a larger, non-symmetric linear system. Given that relationship we then prove a condition on the entries of $J$ that ensures the convergence of the Gaussian Hogwild Gibbs sampling process for any choice of partition or sub-iteration count.

### 4.1.1 A lifting argument and sufficient condition

First observe that we can write multiple steps of Gauss-Seidel as a single step of Gauss-Seidel on a larger system: given $J = L - U$ where $L$ is lower-triangular (including the diagonal, unlike the notation of Section 3) and $U$ is strictly upper-triangular, consider applying Gauss-Seidel to a larger block $k \times k$ system:

$$\begin{pmatrix} L & & -U \\ -U & L & \\ & \ddots & \ddots \\ & & -U & L \end{pmatrix} \xrightarrow{\text{G-S}} \begin{pmatrix} L^{-1} & & \\ L^{-1}UL^{-1} & L^{-1} & \\ \vdots & & \ddots \\ (L^{-1}U)^{k-1}L^{-1} & \cdots & L^{-1}UL^{-1} & L^{-1} \end{pmatrix} \begin{pmatrix} U \\ \\ \\ \end{pmatrix} = \begin{pmatrix} L^{-1}U \\ \vdots \\ (L^{-1}U)^k \end{pmatrix} \quad (6)$$

Therefore one step of Gauss-Seidel on the larger system corresponds to $k$ applications of the Gauss-Seidel update $L^{-1}U$ from the original system to the last block element of the lifted state vector.

Now we provide a lifting on which Gauss-Seidel corresponds to Gaussian Hogwild Gibbs iterations.

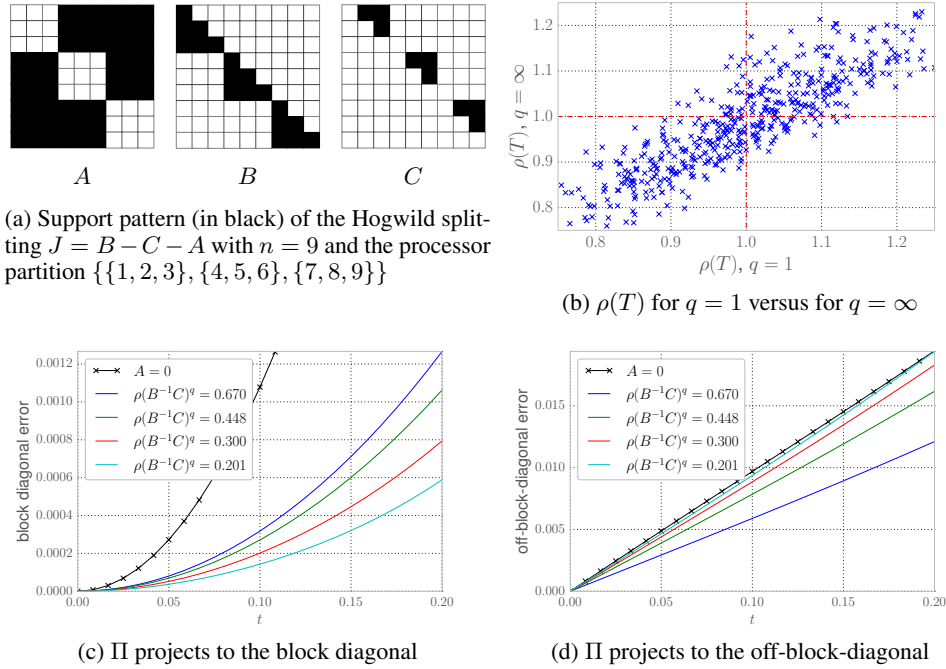

(a) Support pattern (in black) of the Hogwild splitting $J = B - C - A$ with $n = 9$ and the processor partition $\{\{1, 2, 3\}, \{4, 5, 6\}, \{7, 8, 9\}\}$

(b) $\rho(T)$ for $q = 1$ versus for $q = \infty$

(c) $\Pi$ projects to the block diagonal

(d) $\Pi$ projects to the off-block-diagonal

Figure 1: (a) visualization of the Hogwild splitting; (b) Hogwild stability for generic models; (c) and (d) typical plots of $\|\Pi(\Sigma - \Sigma_{\text{hog}})\|_{\text{Fro}}$. In (b) each point corresponds to a sampled model $J = QQ^{\mathsf{T}} + nrI$ with $Q_{ij} \overset{\text{iid}}{\sim} \mathcal{N}(0, 1)$, $r \overset{\text{iid}}{\sim} \text{Uniform}[0.5, 1]$, $n = 24$ with an even partition of size 4. In (c) and (d), models are $J = B - C - tA$ where $B - C - A = QQ^{\mathsf{T}}$, $n = 150$ with an even partition of size 3. The plots can be generated with `python figures.py -seed=0`.

**Proposition 2.** *Two applications of the Hogwild update $T$ of* (5) *are equivalent to the update to the last block element of the state vector in one Gauss-Seidel iteration on the* $(2qn) \times (2qn)$ *system*

$$\begin{pmatrix} E & -F \\ -F & E \end{pmatrix} \tilde{x} = \begin{pmatrix} h \\ \vdots \\ h \end{pmatrix} \text{ with } E = \begin{pmatrix} B \\ -C & B \\ & \ddots & \ddots \\ & & -C & B \end{pmatrix} \quad F = \begin{pmatrix} A+C \\ A \\ \vdots \\ A \end{pmatrix}. \tag{7}$$

*Proof.* By comparing to the block update in (3), it suffices to consider $E^{-1}F$. Furthermore, since the claim concerns the last block entry, we need only consider the last block row of $E^{-1}F$. $E$ is block lower-bidiagonal as the matrix that is inverted in (6), so $E^{-1}$ has the same lower-triangular form as in (6) and the product of the last block row of $E^{-1}$ with the last block column of $F$ yields $(B^{-1}C)^q + \sum_{j=0}^{q-1} (B^{-1}C)^j B^{-1}A = T$. □

**Proposition 3.** *Gaussian Hogwild Gibbs sampling is convergent if Gauss-Seidel converges on* (7).

Unfortunately the lifting is not symmetric and so we cannot impose positive semi-definiteness on the lifted system; however, another sufficient condition for Gauss-Seidel stability can be applied:

**Theorem 1.** *If $J$ is generalized diagonally dominant (i.e. an H-matrix, see Berman et al. [13, Definition 5.13, Theorem 5.14]) then Hogwild Gibbs sampling is convergent for any variable partition and any number of sub-iterations.*

*Proof.* If $J$ is generalized diagonally dominant then there exists a diagonal scaling matrix $R$ such that $\tilde{J} := JR$ is row diagonally dominant, i.e. $\tilde{J}_{ii} \geq \sum_{j \neq i} |\tilde{J}_{ij}|$. Since each scalar row of the coefficient matrix in (7) contains only entries from one row of $J$ and zeros, it is generalized diagonally dominant with a scaling matrix that consists of $2q$ copies of $R$ along the diagonal. Finally, Gauss-Seidel iterations on generalized diagonally dominant systems are convergent [13, Theorem 5.14], so by Proposition 3 the corresponding Hogwild Gibbs iterations are convergent. □

In terms of Gaussian graphical models, generalized diagonally dominant models include tree models and latent tree models (since H-matrices are closed under Schur complements), in which the density of the distribution can be written as a tree-structured set of pairwise potentials over the model variables and a set of latent variables. Latent tree models are useful in modeling data with hierarchical or multi-scaled relationships, and this connection to latent tree structure is evocative of many hierarchical Bayesian models. More broadly, diagonally dominant systems are well-known for their tractability and applicability in many other settings [19], and Gaussian Hogwild Gibbs provides another example of their utility.

Because of the connection to linear system solvers based on two-stage multisplittings, this result can be identified with [18, Theorem 2.3], which shows that if the coefficient matrix is an H-matrix then the two-stage iterative solver is convergent. Indeed, by the connection between solvers and samplers one can prove our Theorem 1 as a corollary to [18, Theorem 2.3] (or vice-versa), though our proof technique is much simpler. The other results on two-stage multisplittings [18, 14], can also be applied immediately for results on the convergence of Gaussian Hogwild Gibbs sampling.

The sufficient condition provided by Theorem 1 is coarse in that it provides convergence for any partition or update schedule. However, given the complexity of the processes, as exhibited in Figure 1b, it is difficult to provide general conditions without taking into account some model structure.

### 4.1.2 Exact local block samples

Convergence analysis simplifies greatly in the case where exact block samples are drawn at each processor because $q$ is sufficiently large or because another exact sampler [9, 10] is used on each processor. This regime of Hogwild Gibbs sampling is particularly interesting because it minimizes communication between processors.

In (4), we see that as $q \to \infty$ we have $T \to T_{\text{block}}$; that is, the deterministic part of the update becomes the block Jacobi update map, which admits a natural sufficient condition for convergence:

**Proposition 4.** *If* $((B - C)^{-\frac{1}{2}} A (B - C)^{-\frac{1}{2}})^2 \prec I$, *then block Hogwild Gibbs sampling converges.*

*Proof.* Since similarity transformations preserve eigenvalues, with $\bar{A} := (B - C)^{-\frac{1}{2}} A (B - C)^{-\frac{1}{2}}$ we have $\rho(T_{\text{block}}) = \rho((B - C)^{\frac{1}{2}} (B - C)^{-1} A (B - C)^{-\frac{1}{2}}) = \rho(\bar{A})$ and since $\bar{A}$ is symmetric $\bar{A}^2 \prec I \Rightarrow \rho(\bar{A}) < 1 \Rightarrow \rho(T_{\text{block}}) < 1$. $\qquad \square$

### 4.2 Variances

Since we can analyze Gaussian Hogwild Gibbs sampling as a linear dynamical system, we can write an expression for the steady-state covariance $\Sigma_{\text{hog}}$ of the process when it is stable. For a general stable LDS of the form $x^{(t+1)} = T x^{(t)} + v^{(t)}$ with $v^{(t)} \sim \mathcal{N}(0, \Sigma_{\text{inj}})$ the steady-state covariance is given by the series $\sum_{t=0}^{\infty} T^t \Sigma_{\text{inj}} T^{t\mathsf{T}}$, which is the solution to the linear discrete-time Lyapunov equation $\Sigma - T \Sigma T^{\mathsf{T}} = \Sigma_{\text{inj}}$ in $\Sigma$.

The injected noise for the outer loop of the Hogwild iterations is generated by the inner loop, which itself has injected noise with covariance $D$, the diagonal of $J$, so for Hogwild sampling we have $\Sigma_{\text{inj}} = \sum_{j=0}^{q-1} (B^{-1} C)^j B^{-1} D B^{-\mathsf{T}} (B^{-1} C)^{j\mathsf{T}}$. The target covariance is $J^{-1} = (B - C - A)^{-1}$.

Composing these expressions we see that the Hogwild covariance is complicated, but we can analyze some salient properties in at least two regimes: when $A$ is small and when local processors draw exact block samples (e.g. when $q \to \infty$).

### 4.2.1 First-order effects in $A$

Intuitively, the Hogwild strategy works best when cross-processor interactions are small, and so it is natural to analyze the case when $A$ is small and we can discard terms that include powers of $A$ beyond first order.

When $A = 0$, the model is independent across processors and both the exact covariance and the Hogwild steady-state covariance for any $q$ is $(B - C)^{-1}$. For small nonzero $A$, we consider $\Sigma_{\text{hog}}(A)$

to be a function of $A$ and linearize around $A = 0$ to write $\Sigma_{\text{hog}}(A) \approx (B - C)^{-1} + [D_0\Sigma_{\text{hog}}](A)$, where the derivative $[D_0\Sigma_{\text{hog}}](A)$ is a matrix determined by the linear equation

$$[D_0\Sigma_{\text{hog}}](A) - S[D_0\Sigma_{\text{hog}}](A)S^{\mathsf{T}} = \widetilde{A} - S\widetilde{A}S^{\mathsf{T}} - (I - S)\widetilde{A}(I - S)^{\mathsf{T}}$$

where $\widetilde{A} := (B - C)^{-1}A(B - C)^{-1}$ and $S := (B^{-1}C)^q$. See the supplementary materials. We can compare this linear approximation to the linear approximation for the exact covariance:

$$J^{-1} = [I + (B - C)^{-1}A + ((B - C)^{-1}A)^2 + \cdots](B - C)^{-1} \approx (B - C)^{-1} + \widetilde{A}. \quad (8)$$

Since $\widetilde{A}$ has zero block-diagonal and $S$ is block-diagonal, we see that to first order $A$ has no effect on the block-diagonal of either the exact covariance or the Hogwild covariance. As shown in Figure 1c, in numerical experiments higher-order terms improve the Hogwild covariance on the block diagonal relative to the $A = 0$ approximation, and the improvements increase with local mixing rates.

The off-block-diagonal first-order term in the Hogwild covariance is nonzero and it depends on the local mixing performed by $S$. In particular, if global synchronization happens infrequently relative to the speed of local sampler mixing (e.g. if $q$ is large), $S \approx 0$ and $D_0\Sigma_{\text{hog}} \approx 0$, so $\Sigma_{\text{hog}} \approx (B - C)^{-1}$ (to first order in $A$) and cross-processor interactions are ignored (though they are still used to compute the correct mean, as per Proposition 1). However, when there are directions in which $S$ is slow to mix, $D_0\Sigma_{\text{hog}}$ picks up some parts of the correct covariance's first-order term, $\widetilde{A}$. Figure 1d shows the off-block-diagonal error increasing with faster local mixing for small $A$.

Intuitively, more local mixing, and hence relatively less frequent global synchronization, degrades the Hogwild approximation of the cross-processor covariances. Such an effect may be undesirable because increased local mixing reflects greater parallelism (or an application of more powerful local samplers [9, 10]). In the next subsection we show that this case admits a special analysis and even an inexpensive correction to recover asymptotically unbiased estimates for the full covariance matrix.

### 4.2.2   Exact local block samples

As local mixing increases, e.g. as $q \to \infty$ or if we use an exact block local sampler between global synchronizations, we are effectively sampling in the lifted model of Eq. (3) and therefore we can use the lifting construction to analyze the error in variances:

**Proposition 5.** *When local block samples are exact, the Hogwild sampled covariance $\Sigma_{Hog}$ satisfies*

$$\Sigma = (I + (B - C)^{-1}A)\Sigma_{Hog} \quad and \quad ||\Sigma - \Sigma_{Hog}|| \leq ||(B - C)^{-1}A|| \, ||\Sigma_{Hog}||$$

*where $\Sigma = J^{-1}$ is the exact target covariance and $||\cdot||$ is any submultiplicative matrix norm.*

*Proof.* Using the lifting in (3), the Hogwild process steady-state covariance is the marginal covariance of half of the lifted state vector, so using Schur complements we can write $\Sigma_{\text{Hog}} = ((B - C) - A(B - C)^{-1}A)^{-1} = [I + ((B - C)^{-1}A)^2 + \cdots](B - C)^{-1}$. We can compare this series to the exact expansion in (8) to see that $\Sigma_{\text{Hog}}$ includes exactly the even powers (due to the block-bipartite lifting), so therefore $\Sigma - \Sigma_{\text{Hog}} = [(B - C)^{-1}A + ((B - C)^{-1}A)^3 + \cdots](B - C)^{-1} = (B - C)^{-1}A\Sigma_{\text{Hog}}$.   □

## 5   Conclusion

We have introduced a framework for understanding Gaussian Hogwild Gibbs sampling and shown some results on the stability and errors of the algorithm, including (1) quantitative descriptions for when a Gaussian model is not "too dependent" to cause Hogwild sampling to be unstable (Proposition 2, Theorem 1, Proposition 4); (2) given stability, the asymptotic Hogwild mean is always correct (Proposition 1); (3) in the linearized regime with small cross-processor interactions, there is a tradeoff between local mixing and error in Hogwild cross-processor covariances (Section 4.2.1); and (4) when local samplers are run to convergence we can bound the error in the Hogwild variances and even efficiently correct estimates of the full covariance (Proposition 5). We hope these ideas may be extended to provide further insight into Hogwild Gibbs sampling, in the Gaussian case and beyond.

## 6   Acknowledgements

This research was supported in part under AFOSR Grant FA9550-12-1-0287.

## Footnotes

[1]Assume models are non-degenerate: matrix parameters are of full rank and densities are finite everywhere.

[2] When $J$ is symmetric one can arrive at the same condition by applying a similarity transform as in Proposition 5. We use the lifting argument here because we extend the idea in our other proofs.

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
