[Supplementary Material · nips2013_supplemental.pdf]

# Supplementary Materials

Revised November 8, 2013

## 1 Derivation of Covariance Derivative Expression

Writing explicit dependence on $A$ and with $S := (B^{-1}C)^q$ and $T(A) = S + (I-S)(B-C)^{-1}A$ as in the text, we have

$$\Sigma_{\text{hog}}(A) = T(A)\ \Sigma_{\text{hog}}(A)\ T^{\mathsf{T}}(A) + \widetilde{D} \tag{1}$$

from the discrete-time Lyapunov equation given in the text at the start of Section 4.2, where $\widetilde{D} = \sum_{j=0}^{q-1}(B^{-1}C)^j B^{-1}DB^{-\mathsf{T}}(B^{-1}C)^{q\mathsf{T}}$. Taking the total derivative of both sides with respect to $A$ and evaluating at $0$ we have

$$[D_0\Sigma_{\text{hog}}](A) - T(0)\ [D_0\Sigma_{\text{hog}}](A)\ T^{\mathsf{T}}(0) = [D_0T](A)\ \Sigma_{\text{hog}}(0)\ T^{\mathsf{T}}(0) + T(0)\ \Sigma_{\text{hog}}(0)\ [D_0T]^{\mathsf{T}}(A) \tag{2}$$

where $[D_0T](A) = (I-S)(B-C)^{-1}A$. Substituting $T(0) = S$ and $\widetilde{A} := (B-C)^{-1}A(B-C)^{-1}$ we have

$$[D_0\Sigma_{\text{hog}}](A) - S\ [D_0\Sigma_{\text{hog}}](A)\ S^{\mathsf{T}} = (I-S)\widetilde{A}S^{\mathsf{T}} + S\widetilde{A}(I-S)^{\mathsf{T}}. \tag{3}$$