[Reviews · NeurIPS 2013]

Submitted by Assigned_Reviewer_5

This paper relates classic results from parallel linear system solvers and linear dynamical systems to the analysis of parallel Gibbs sampling using a Hogwild update strategy which violates the Markov blanket independence assumptions. By relating the Hogwild Gaussian Gibbs sampler to these classic results the authors are able to make strong statements about the structure of the posterior distribution both in terms of its stability as well as the resulting mean and covariance.

The paper is dense but well written.

What I like most about this paper is how it really emphasizes the connection between parallel Gibbs sampling and the classic Gauss-Seidel and Jacobi linear solvers and how this connection can lead to new insight. I believe this connection and the splitting strategy could be useful in the broader design and analysis of parallel algorithm in ML.


Minor comments:

After the proof of Theorem 1 there is some discussion suggesting that the Hogwild Gibbs sampler would perform best in tree structured models (or diagonally dominant models). However, in these models wouldn't an ergodic parallel sampling strategy perform well (e.g., the Chromatic Gibbs Sampler since trees are two colorable).

Labeling and directly referencing the subparts of figure 1a would help a lot since the splitting can be a bit confusing.

Summary: This well written paper connects parallel Gibbs sampling with classic results in linear systems solvers and linear dynamical systems and uses these connections to analyze the Hogwild Gibbs sampler when applied to Gaussian graphical models. These connections could have a significant impact in how we design and analyze parallel Gibbs samplers.

Submitted by Assigned_Reviewer_7

The authors perform an analysis of "Hogwild" parallel Gibbs sampling for Gaussian distributions and show a connection between Gauss-Seidel / Jacobi and the Hogwild routine. They exploit this connection to show conditions for when this parallel Gibbs sampling process converges to the correct mean, and they are also able to make statements about the covariances of the system.

I enjoyed reading the connection between Gauss-Seidel and Gauss-Jacobi and parallel Gaussian Gibbs sampling and find that this type of analysis is very useful for the NIPS community as parallel Gibbs sampling has received relatively little theoretical attention. A few comments:

1) I guess for the simple case of Gaussians, parallel Gibbs sampling is overkill as one can just directly obtain samples quickly for any multivariate Gaussian (but of course it is useful for analysis purposes). It would be nice to point this out (not sure if I agree with the sentiment in line 69) and also to make a few statements about how this analysis could also be applied to non-Gaussian cases.

2) I have a technical question about the role of the v vector (samples from the Normal). It seems that for a given iteration t, the same set of v samples would have to be globally shared across all processors for this method to work (e.g., Eq. 4)? In other words, does processor 2 have to know the exact v samples that processor 1 used in order for this Jacobi process to work? If that is the case, then parallel sampling is not really parallel but predetermined in advance based on the global v vector. I would appreciate it if the authors could clarify this issue.

3) It would be nice to have more intuitive explanations after some of the equations (e.g., after proposition 5).

4) Figure 1(b) was somewhat surprising as intuitively it seems the processors would locally drift more when q=infinity which would hurt the global model more. It might be worth taking a look at the dissertation of Asuncion (on of the co-authors of AD-LDA) who analyzed the Markov chain transition matrix of AD-LDA for a simple case and found that the stationary distribution of AD-LDA got worse when local processors were allowed to fully converge.
Summary: The authors perform an analysis of "Hogwild" parallel Gibbs sampling for Gaussian distributions and show a connection between Gauss-Seidel / Jacobi and the Hogwild routine. They exploit this connection to show conditions for when this parallel Gibbs sampling process converges to the correct mean, and they are also able to make statements about the covariances of the system. I enjoyed reading the connection between Gauss-Seidel and Gauss-Jacobi and parallel Gaussian Gibbs sampling and find that this type of analysis is very useful for the NIPS community as parallel Gibbs sampling has received relatively little theoretical attention.

Submitted by Assigned_Reviewer_8

The authors present analysis of "hogwild" parallel Gibbs sampling, where Gibbs updates are run for blocks of variables separately on each processor without communicating each iteration, so that the sampler is no longer "correct" at least in the traditional sense. The analysis focuses on sampling Gaussian graphical models, with the main result being that under a condition on the precision matrix (being generalised diagonally dominant) hogwild Gaussian Gibbs sampling is stable, and gives the correct asympoptic mean. The story for variances is a little more complex: a finite error is introduced when there are cross processor interactions that are ignored, and these become more important if the mixing at each processor is faster, but in the limit of drawing exact samples at each processor the variance can be corrected analytically.

Quality. The ideas and analysis are novel and rigorous as far as I can tell.

Clarity. The paper is reasonably well written but could be better organised, especially for the typically ML reader who is probably not overly familar with the linear algebra terms or the various linear solvers introduced. Having a "background material" section first laying this groundwork might be helpful.

Originality. I am not aware of other analysis of this type of sampling.

Significance. Scaling sampling methods is of significant interest to the NIPS community, so analysis of methods that attempt to do this is valuable, especially if it points the way towards algorithmic improvements. This paper laid out some nice ideas, many of which seem intuitive: hogwild sampling is dependent on the cross processor interactions not being too strong, and faster local mixing might actually be detrimental in some ways. However, the connection of these ideas to potential algorithm improvements was only hypothetical and then only hinted at. The paper begs the question in several cases. The generalised diagonally dominant condition in practice means I should try to put highly interacted variables: how should I try to do this? How well does it work in practice? Proposition 5 gives an analytic correct for the variance when q=inf: what happens if I use this for q large? I think this would have been a stronger paper if these questions had been addressed: if not empirically, at least in the discussion. But I accept there is a space limitation.

A relevant reference that also used "hogwild" sampling is
Finale Doshi-Velez, Shakir Mohamed, David A. Knowles and Zoubin Ghahramani. Large scale nonparametric Bayesian inference: Data parallelisation in the Indian buffet process. NIPS 2009.

Minor notes
Equation 1: am I missing something or there should be a minus sign?!
Next equation (line 141): hat over v

Summary: A nice piece of analysis that could have been more clearly laid out for the non-expert, and would have benefitted from more discussion (or experiments) about the practical implications.
Author Feedback

Author rebuttal: # Overview #

We thank the reviewers for their thoughtful comments and suggestions.

As we understand the reviews, the reviewers agreed that the work is interesting, original, and potentially impactful. The main criticisms were that the paper is dense and that it could comment on a few more points or in greater detail. Given the page limit there is a degree of tension between these two objectives, but we can make some revisions to clarify points in the paper and respond in greater detail here.

# Assigned_Reviewer_5 #

Indeed, tree models can be sampled exactly with message passing in time linear in the number of nodes. However, our point is that generalized diagonally dominant models include as special cases both tree models and *latent* tree models. Latent tree models may in fact be fully connected; there need only exist some (unknown) set of extra latent variables that could be introduced to yield a tree. Latent tree models are used to model multiresolution or hierarchical long-range effects and they are of independent interest, so we point out that they are included in the class of models for which we prove Gaussian Hogwild Gibbs sampling is convergent. This result also suggests that these samplers may be effective more generally for many models which have multiresolution or hierarchical structure.

# Assigned_Reviewer_7 #

The Gaussian case is certainly interesting for analysis of machine learning methods, but Gaussian sampling and inference algorithms are an active area of research both in machine learning and in other areas. Drawing exact samples with a direct method (such as a Cholesky decomposition) has cubic complexity in the number of variables, and for large models, such as in Gaussian Process models or large Markov random fields, cubic complexity can be too expensive.

To emphasize that Gaussian sampling and other inference methods are of interest not only as an analytical tool but also as a practical one, we will add appropriate references, such as

"Efficient sampling for Gaussian process inference using control variables" in Advances in Neural Information Processing Systems. 2008.


By construction, the Hogwild Gibbs sampling algorithm does not communicate the local samples during the inner loop, and that is reflected in Eq. 4 by the fact that v^{(j,t)} is diagonally-shaped noise, i.e. it is a collection of independent scalar Gaussians. Further, note that because B and C are block diagonal matrices, the only place where block entries of the state vector x^{(t)} affect one another is when A is applied, which happens once per outer iteration when the statistics are synchronized.


We thank the reviewer for the reference, and we think there may be an interesting connection to be made at some level: as we show in our first-order analysis in Section 4.2.1 and in the simulation in Figure 1(d), when processors are allowed to fully mix the cross-processor variances estimates are made worse (while means remain correct).

# Assigned_Reviewer_8 #

We agree that there are many more questions to ask and perhaps answer with our proposed framework, but given space constraints and that the paper is already agreed to be dense, we did our best to lay out both the framework and the results for which we did have room concerning convergence, correctness of means, and errors (with tradeoffs) and corrections in variances.


The question of how best to organize the variables is very important for practical considerations, as the reviewer points out. Exploring several methods with empirical comparisons, as the reviewer suggests, and perhaps analysis using some of the tools we develop here may require its own paper.


Our framework does provide a way to analyze the error when applying the asymptotic correction given that local samplers have not fully mixed, though we do not have space to include such an analysis in this paper. As a brief sketch, the local process covariance on the kth processor converges (in any submultiplicative matrix norm, such as Frobenius norm) to its asymptotic value at a rate of rho(B_k^{-1} C_k)^2 per inner iteration. Further, from the linear dynamical system analysis in the paper it is clear that to analyze the effect of any error in the local covariances on the estimated overall covariance, one needs only to analyze the perturbation of a linear system (the outer iterations' discrete time Lyapunov system) and the scaling by the linear correction factor. There are many relevant results on bounding errors for such discrete-time Lyapunov systems in "Sensitivity of the solutions of some matrix equations" by Rajendra Bhatia and Ludwig Elsner (1997). Qualitatively, our framework shows that if the local covariance error (which decreases at least as fast as rho(B^-1 C)^(2q) with q) is small, the error in the corrected covariance estimates is also small.

We do not believe this paper has room for an appropriate treatment of this issue, though we hope we have demonstrated here that it is readily approachable with the framework.


There is indeed a missing minus sign in Eq. 1; it was a typesetting mistake and does not affect any other expressions in the paper.